# Preparation of AlON Powder by Carbothermal Reduction and Nitridation with Assisting by Silane Coupling Agent

**DOI:** 10.3390/ma16041495

**Published:** 2023-02-10

**Authors:** Zhongyuan Xue, Xingming Wang, Yuyang Liu, Xue Bai, Tao Gui, Xingqi Wang, Xiaoning Li

**Affiliations:** 1National Engineering Research Center of Environment-Friendly Metallurgy in Producing Premium Non-Ferrous Metals, China GRINM Group Corporation Limited, Beijing 100088, China; 2GRINM Resources and Environment Tech. Co., Ltd., Beijing 101407, China; 3General Research Institute for Non-Ferrous Metals, Beijing 100088, China; 4Beijing Engineering Research Center of Strategic Nonferrous Metals Green Manufacturing Technology, Beijing 101407, China

**Keywords:** carbothermic reduction and nitridation, distribution uniformity of raw powders, silane coupling agent, AlON

## Abstract

In the preparation processes of aluminum oxynitride (AlON) powders by carbothermal reduction and nitridation, the homogeneity of mixed raw powders between Al_2_O_3_ and C is a critical factor by which the final composition and related properties of AlON transparent ceramic will be decided. In this paper, a silane coupling agent was used as a dispersant to optimize the distribution uniformity of raw material of Al_2_O_3_ and C, and the preparation of AlON powder with controllable composition and its distribution is investigated. The results show that the silane dispersant could effectively improve the distribution uniformity of raw material. The silane coupling agent contains functional groups of −SiH_3_ and −C_n_H_2n+1_O. XPS showed that the silane could react with C and Al_2_O_3_ to form the Si–C bond and C–Al_2_O_3_ bond, respectively. The silane coupling agent provides a connected bridge for raw material powders. When the amount of the silane was 5 wt%, the mixed powder had a great distribution uniformity. The addition of silane coupling agent improved the reactivity of raw materials and decreased the synthesis temperature of AlON. The single-phase AlON powder was obtained after the Al_2_O_3_/C mixed powder was kept at 1670 °C for 30 min. Furthermore, the grain size of AlON powder was 100–200 nm with an AlN content of 27.5 mol%. With the increase of holding time to 4 h, the grain size increased to 15 μm, indicating that sintering between particles occurred, which may reduce the sintering activity of the powder.

## 1. Introduction

AlON transparent ceramics have excellent mechanical properties and optical properties, which are often used as the window material for transparent armor and infrared fairings [1,2]. The properties of AlON ceramics depend on the quality of the powder. The synthesis methods of AlON powder mainly include aluminothermic reduction and nitridation (ARN) [3,4], solid-state reaction (SSR) [5,6], and carbothermal reduction and nitridation (CRN) [7,8,9,10], among which carbothermal reduction and nitridation have attracted much attention due to the low cost and easy industrialization. [8,10,11,12,13].

The reaction process of AlON powder synthesis mainly includes three stages: AlN generation, O-enriched AlON generation, and N-enriched AlON generation [14,15]. The stage of AlN generation is the most important. In the preparation of AlON powders by carbothermal reduction and nitridation, the nitrogen content of AlON ceramics can be accurately controlled by the ratio of Al_2_O_3_ and C. However, the final composition of AlON powder can be fluctuating, due to the insufficient carbothermal reduction and nitridation reaction caused by the severe agglomeration in the raw material. The AlN content of AlON transparent ceramics is ideally between 21.6 mol% and 29.7 mol%, corresponding with different optical properties. When the AlN content is 27.5 mol%, AlON transparent ceramics had the best optical properties [16]. Therefore, the control of AlN content, which is dominated by the distribution of raw materials, is of great significance in improving the properties of AlON transparent ceramics. Recently, the preparation of precursors with core–shell structures to improve the uniformity of raw material distribution has attracted much attention [8,9,12,13]. There are few studies on improving the distribution uniformity of raw materials by adding dispersants. Comparatively, the utilization of dispersant can be more easily commercialized.

Generally, the agglomeration of Al_2_O_3_ powder was adjusted by adding modifiers, such as polyethylene glycol [17], polyethylene imine [18], silane coupling agent [19,20], and so on. Specially, the silane coupling agent is attracting more and more attention because the Si functional groups in silane can form covalent bonds with metals, metal oxides, silicates, and other inorganic substances [21,22,23]. Therefore, in this paper, the silane coupling agent was used as dispersant, in order to improve the uniformity of Al_2_O_3_/C mixed powder. The effect of silane coupling agent content on the uniformity of Al_2_O_3_/C mixed powder was analyzed. Moreover, the mechanism of action between the silane coupling agent and raw material powders was revealed. Further, the preparation conditions of carbothermal reduction and nitridation, including the holding time and synthesis temperature, were investigated. The single-phase AlON was obtained at the reaction temperature of 1670 °C with the holding time of 30 min. The powder particle size was between 100 and 200 nm with an AlN content of 27.5 mol%. This study has good prospects for the application of industrialization of preparation of AlON powder by the method of carbothermal reduction and nitridation.

## 2. Materials and Methods

γ–Al_2_O_3_ (>99.99%, 40 nm, Beijing Deke Daojin Science and Technology Co., Ltd., Beijing, China), activated carbon (AR, 2.5 μm, Sinopharm Chemical Reagent Co. Ltd., Shanghai, China) and silane coupling agent (>99%, alkoxysilane, C_n_H_2n+1_OSiH_3_, Hubei Jianghan New Materials Co., Ltd., Jingzhou, China) were used in this work.

The γ–Al_2_O_3_ and C (5.8 wt%) were mixed by ball milling in Teflon jars. With anhydrous ethanol as the ball milling medium, alumina balls as the grinding ball, the silane coupling agent was added into the Al_2_O_3_/C mixed slurry as the dispersing agent for ball milling of 24 h. In the process of ball milling, the ratio of balls to powder was 7:1, the ratio of ethanol to powder was 1.4:1, and the rotation rate remained at 50 rpm. The slurry was dried and screened through a 300 mesh nylon sieve. The obtained powder was placed in a BN crucible, and AlON (Al_22.53_O_28.41_N_3.59_, AlN mol% = 27.5) powder was synthesized at 1640–1730 °C in a graphite furnace under a flowing nitrogen atmosphere. In order to ensure the repeatability of the experimental results, all the experiments were carried out three times.

The phase of the synthesized powder was analyzed by X-ray diffractometer (XRD, D/max 2500, RIGAKU, Matsumoto, Japan). The composition and microstructure of the powder were detected by scanning electron microscopy (SEM, JSM-F100, JEOL, Osaka, Japan) equipped with energy dispersive spectroscopy (EDS). SEM was also used to analyze the grain size of AlON powder. The thermal analysis of the powder was carried out by a synchronous thermal analyzer (TG–DSC, STA449F1, NETZSCH, Selb, Germany). The elemental composition and chemical state of the powder were tested by X-ray photoelectron spectroscopy (XPS, Nexsa G2, Thermo Fisher Scientific, Waltham, MA, USA).

## 3. Results

### 3.1. Effect of Dispersant Content on the Dispersion Uniformity of Raw Materials

The SEM and EDS results of Al_2_O_3_/C mixed powders with different dispersant contents are shown in Figure 1. When dispersant was not added, the distribution of C was uneven, and the size of soft agglomerated particles exceeded 30 μm. When the contents of dispersant were 5 wt% and 7.5 wt%, the distribution of C and O elements tended to be more uniform. However, when the content of the dispersant was 7.5 wt% and 10 wt%, the agglomeration of raw powder was severe. The size of some particles exceeded 50 μm. Therefore, when the additional amount of dispersant was 5 wt%, the distribution uniformity of C and O elements can be improved and the abnormal agglomeration of the raw material was inhibited.

### 3.2. Interaction between the Silane Coupling Agent and Raw Materials

In order to clarify the action mechanism of dispersant in Al_2_O_3_/C mixed powder, the element status of the mixed powder was analyzed by XPS. Figure 2 shows the C1s spectrum of Al_2_O_3_/C mixed powder with different contents of dispersants, and the XPS spectrum of mixed powder with different contents of dispersants is displayed in Figure 3. As shown in Figure 2, all mixed powders have a peak at 284.6 eV. Combined with the Si2p peak corresponding to the binding energy of 102 eV in Figure 3, the peak with the binding energy of 284.6 eV in the C1s spectrum belonged to the C–Si bond [24]. The Si functional group in the silane coupling agent interacted with C to form Si–C bond. When the content of the dispersant increased from 2.5 wt% to 5 wt%, the O–C–O bond in the C functional group broke and then formed a C–Al_2_O_3_ bond with Al_2_O_3_. When the content of the dispersant was 10 wt%, Si–C bonds and C–O bonds were found in the C1s spectrum, the interaction between the Si functional group of dispersant and C was weakened, and the interaction between powders was enhanced. The integral calculation of peak area shows that the proportion of Si–C bond was 83.01%, 94.80%, 89.16%, and 57.86%, respectively, with the content of dispersant increased from 2.5 wt% to 10 wt%. When the content of the dispersant was 5 wt%, the proportion of the C–Si bond was the highest. At this time, the raw material uniformity of mixed powder was the best.

Figure 4 shows the action mechanism of the dispersant in the ball milling process. The carbon would adhere to the surface of Al_2_O_3_ particles after ball milling [7]. Al_2_O_3_ has a large specific surface area, which can easily agglomerate during the ball milling process. At this time, the distribution of carbon on the surface will be uneven, and the agglomerated powder will reduce the reaction activity. After adding the dispersant to the mixed powder, the Si functional group and C functional group in the dispersant interact with C and Al_2_O_3_ in the mixed powder, respectively. The interaction of powder particles will be weakened. At the same time, the dispersant is uniformly dispersed between C and Al_2_O_3_, and the distribution uniformity of raw material was increased.

### 3.3. Carbothermal Reduction and Nitridation Synthesis of AlON Powder Assisted by Silane Coupling Agent

#### 3.3.1. Effect of Synthesis Temperature on the Phase Composition of Reaction Products

Figure 5 displays the TG–DSC curves of the Al_2_O_3_/C mixed powder with different contents of dispersants in a nitrogen atmosphere. A broad exothermic peak appeared at about 600 °C due to the change in the lattice type of γ–Al_2_O_3_ [25]. The reaction formula is as follows:γ–Al_2_O_3_ → δ–Al_2_O_3_ → θ–Al_2_O_3_ → α–Al_2_O_3_(1)

As shown in the TG curves in Figure 5, the mass of mixed powder decreased at the beginning of heating, which was caused by the volatilization of crystal water in γ–Al_2_O_3_. When the heating temperature was 200–400 °C, the mass of the mixed powder with 5 wt% dispersant decreased significantly, which was due to the decomposition of the silane coupling agent in this temperature range. The mass reduction speed accelerated when the temperature rose to 1300 °C. It was attributed to a gas-solid reaction in Al_2_O_3_, which generated CO and caused mass reduction [26]. The corresponding equation is as follows:Al_2_O_3_ + 3C + N_2_ = 2AlN + 3CO(2)

The DSC curves in Figure 5a,b had exothermic peaks at about 1400 °C. However, the DSC curve with 5 wt% content silane displays severe thermal effect, as shown in Figure 5b. The reaction of mixed raw materials with dispersant was more violent. Therefore, the mixed powder with dispersant has stronger reactivity.

The XRD patterns of the mixed powder with 5 wt% dispersant after reacting at different temperatures for 1 h are shown in Figure 6. After heating at 1640 °C for 1 h, the main phase of the mixed power had been transformed into spinel AlON (γ–AlON), but a few α–Al_2_O_3_ phases still existed. As the temperature increased, the diffraction peak intensity of γ–AlON improved. When the heating temperature exceeded 1670 °C, the α–Al_2_O_3_ phase disappeared, and the reaction product was single phase γ–AlON.

The XRD pattern of the mixed powder with 0 wt% and 5 wt% dispersants held at 1670 °C for 1 h is shown in Figure 7. The reaction product of mixed powder without dispersant after reacting for 1 h at 1670 °C was α–Al_2_O_3_ and γ–AlON. When the content of the dispersant was 5 wt%, the intensity of γ–AlON phase diffraction peak increased and α–Al_2_O_3_ disappeared, and the reaction product was single-phase γ–AlON. Therefore, adding dispersant can effectively promote the synthesis of AlON powder. The comparison of AlON powder synthesis temperature is shown in Table 1. The AlON synthesis temperature of this work is lower than that of other work, which is valuable for low-cost production.

#### 3.3.2. Effect of Reaction Time on Phase Composition and Micro Morphology of Reaction Products

As shown in Figure 8, when the mixed powder (5 wt% dispersant) was heated at 1670 °C for 10 min, the reaction product was α–Al_2_O_3_, AlN, and γ–AlON mixture. When the mixed powder was held for 20 min, the diffraction peak intensity of the α–Al_2_O_3_ phase decreased, and the peaks of γ–AlON initially appeared. When the reaction time extended to more than 30 min, the reaction product was single-phase γ–AlON.

Figure 9 presents the SEM micrographs of AlON powder synthesized at 1670 °C with different holding time. When the holding time was 30 min (Figure 9a), the size of the powder particles was about 100–200 nm. With the holding time increasing from 1 h to 4 h, the connection between grains was more and more obvious, and the steps of grains were increased to form new planes. When the holding time was 4 h, the interconnected grain size was sharply increased to 1–5 μm, and the boundary between the grains disappeared. The interconnection of grains and the appearance of sintering necks and steps were related to grain growth and coarseness, which possibly reduced the specific surface area and the sintering activity of powder. It was not conducive to preparing transparent ceramics in the later stage [23,30]. In this experiment, the uniformity and reactivity of raw material were increased by adding dispersant. The mixed powder was reacted at 1670 °C for 30 min, and the AlON powder with fine dispersed grains was synthesized. Due to the low reaction temperature, the driving force of AlON grains growth was limited, the dispersed grains were still fine.

The fitting calculation was carried out for XRD patterns in Figure 8, and the lattice constant and AlN content of the powder are shown in Table 2. The lattice constant of AlON with a holding time of 30 min was 7.9456 Å, and the lattice parameter of AlON with a holding time of 4 h was 7.9443 Å. The relationship between AlN content and theoretical density, lattice parameter is summarized in Figure 10. As shown in Figure 10, when the content of AlN was 22–34 mol%, the lattice parameter and theoretical density of γ–AlON increased with the increase of AlN content, which caused a change in the refractive index of AlON. Therefore, when the nitrogen content in AlON was different, the light would scatter when passing through the grain boundary, thus affecting the optical properties of ceramics [16,27]. The lattice parameters of γ–AlON were all about 7.944 Å with different holding times, and the AlN content was about 27.5 mol%. The carbothermal reduction and nitridation reaction was complete, and the AlN content of AlON was precisely controlled.

## 4. Conclusions

In this work, the silane coupling agent was used to improve the distribution uniformity of raw material, and the mechanism of action between the silane coupling agent and raw material powders was reclarified. Furthermore, the effect of the silane coupling agent on synthesis of AlON powder was studied. The conclusions are as follows:

(1) After adding the silane coupling agent, Si functional groups and C functional groups bonded with carbon and Al_2_O_3_, respectively, to form Si–C bonds and C–Al_2_O_3_ bonds, which built a bridge between the raw material powder C/Al_2_O_3_ and weakened the interaction between raw materials.

(2) When the amount of silane coupling agent was 5 wt%, the distribution uniformity of C and O elements was effectively improved, and the soft agglomeration of mixed powder was reduced.

(3) The AlON synthesis process assisted by silane coupling agent was optimized, and single-phase AlON was obtained after the reaction at 1670 °C for 30 min. The powder particle size was between 100 and 200 nm with an AlN content of 27.5 mol%.

## Figures and Tables

**Figure 1 materials-16-01495-f001:**
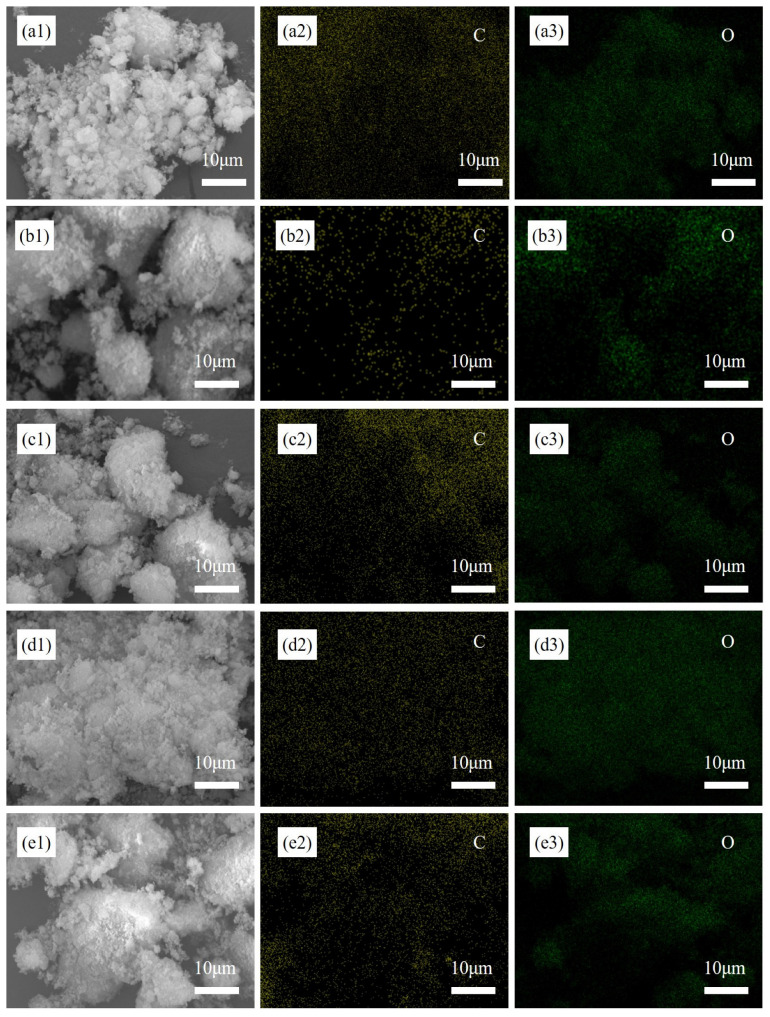
The SEM images and corresponded EDS mapping of Al_2_O_3_/C mixed powder with different contents of silane: (**a1**–**a3**) 0 wt%; (**b1**–**b3**) 2.5 wt%; (**c1**–**c3**) 5 wt%; (**d1**–**d3**) 7.5 wt%; (**e1**–**e3**) 10 wt%.

**Figure 2 materials-16-01495-f002:**
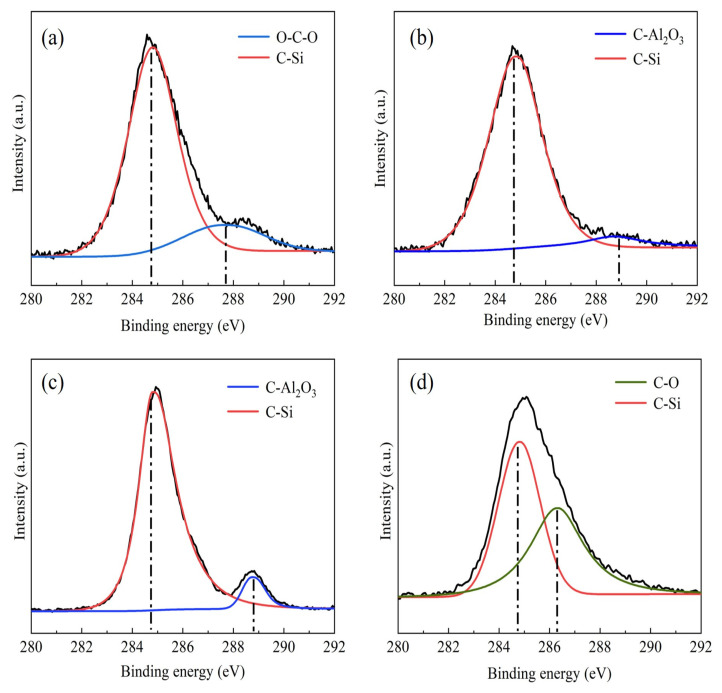
C 1s XPS spectra of Al_2_O_3_/C mixed powder with different contents of dispersants: (**a**) 2.5 wt%, (**b**) 5 wt%, (**c**) 7.5 wt%, (**d**) 10 wt%.

**Figure 3 materials-16-01495-f003:**
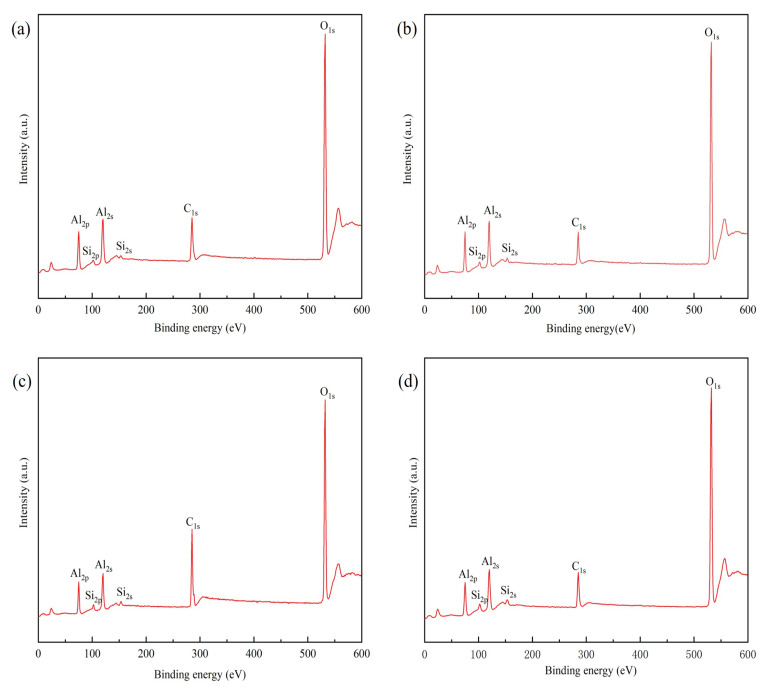
XPS full-survey spectra of Al_2_O_3_/C mixed powder with different contents of dispersants: (**a**) 2.5 wt%, (**b**) 5 wt%, (**c**) 7.5 wt%, (**d**) 10 wt%.

**Figure 4 materials-16-01495-f004:**
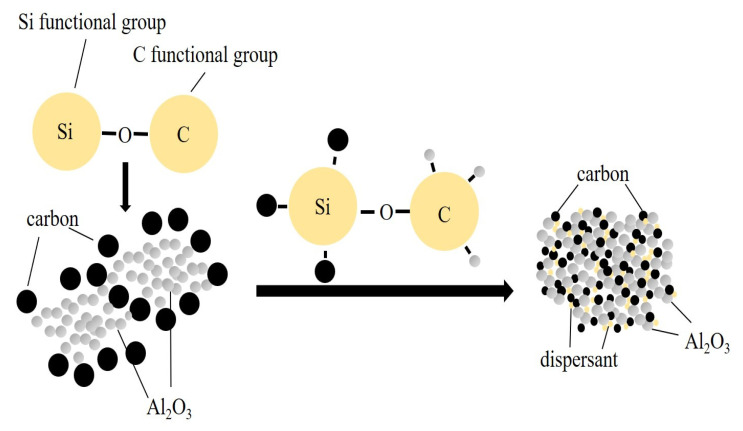
Schematic diagram of the interaction between dispersant and raw powder.

**Figure 5 materials-16-01495-f005:**
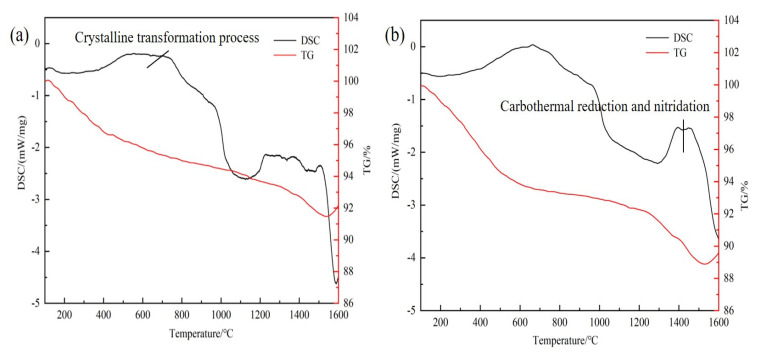
TG–DSC curves of the Al_2_O_3_/C mixed powder with different contents of dispersants in nitrogen: (**a**) 0 wt%; (**b**) 5 wt%.

**Figure 6 materials-16-01495-f006:**
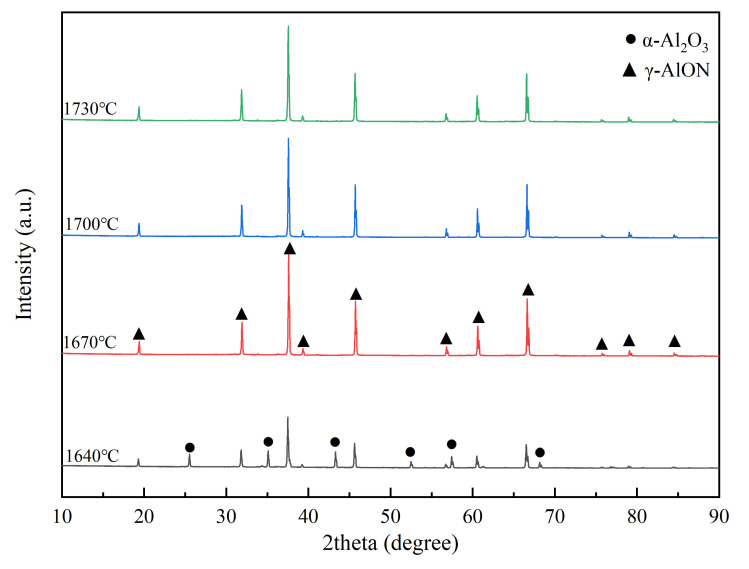
XRD patterns of mixed powders (5 wt% dispersant) after heating for 1 h at different temperatures.

**Figure 7 materials-16-01495-f007:**
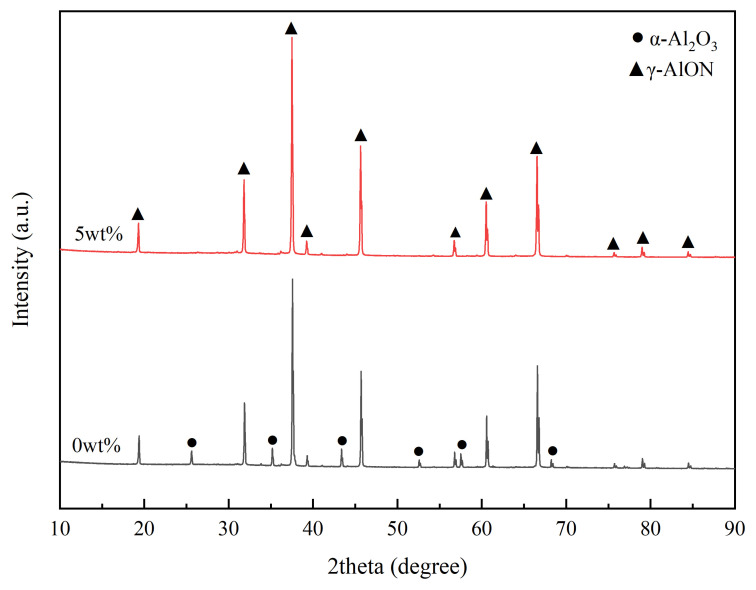
XRD patterns of mixed powder (0 wt% and 5 wt% dispersants) after heating for 1 h at 1670 °C.

**Figure 8 materials-16-01495-f008:**
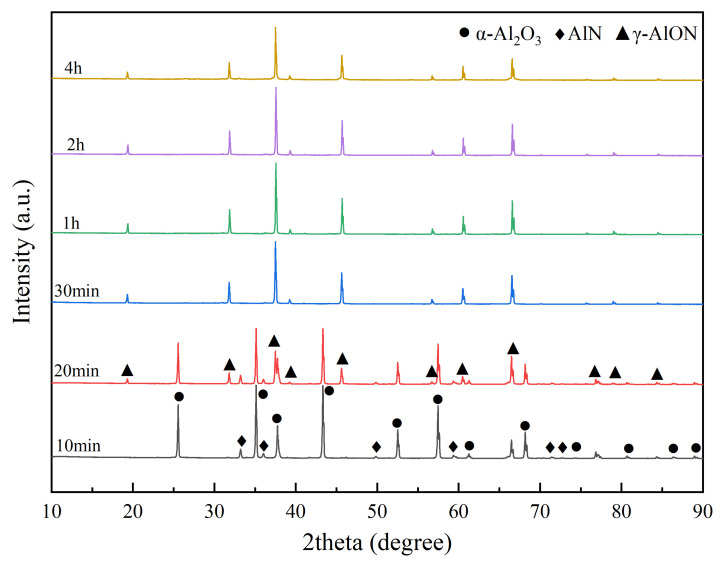
XRD patterns of mixed powders with 5 wt% dispersant at 1670 °C at different time.

**Figure 9 materials-16-01495-f009:**
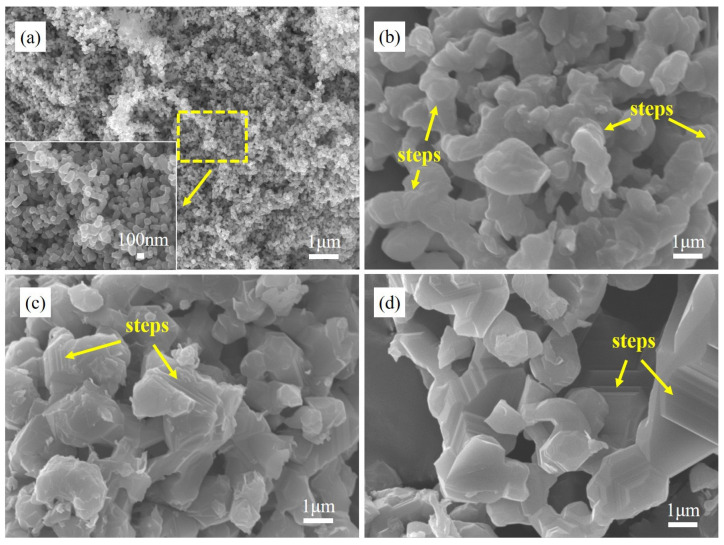
SEM morphologies of AlON synthesized at 1670 °C for various time: (**a**) 30 min; (**b**) 1 h; (**c**) 2 h; (**d**) 4 h.

**Figure 10 materials-16-01495-f010:**
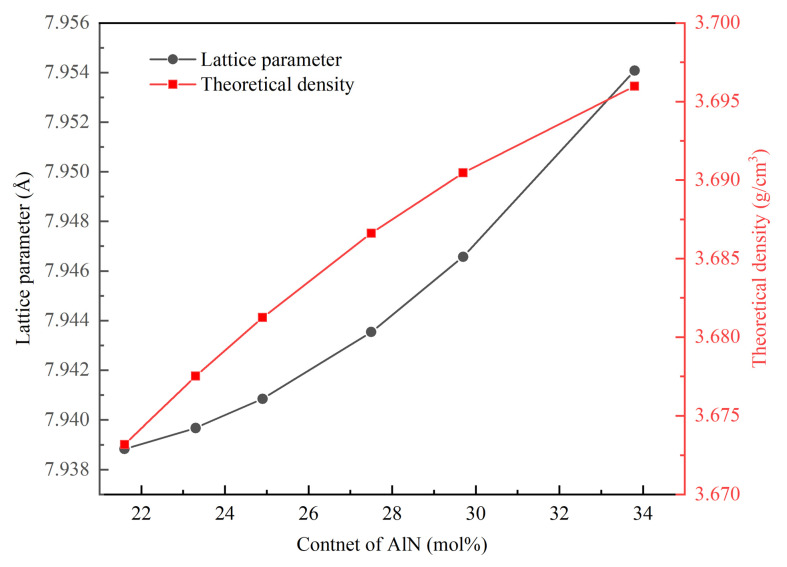
Effect of AlN content on lattice parameter and theoretical density of AlON [16,27,29].

**Table 1 materials-16-01495-t001:** Comparison of synthetic parameters of AlON powder.

Synthesis Method	Holding Temperature (°C)	Reference
CRN	1780	[27]
CRN	1750	[12]
CRN	1750	[28]
CRN	1750	[7]
CRN	1750	[29]
CRN	1750	[13]
CRN	1700	[11]
CRN	1700	[8]
CRN	1670	This work

**Table 2 materials-16-01495-t002:** Lattice parameters, AlN content and theoretical density of AlON synthesized at 1670 °C for various time.

Temperature (°C)	Time (h)	Lattice Parameter (Å)	AlN Content (mol%)	Theoretical Density (g/cm^3^)
1670	0.5	7.9456	27.5	3.68
1670	1	7.9435	27.4	3.68
1670	2	7.9447	27.4	3.68
1670	4	7.9443	27.4	3.68

## Data Availability

The data presented in this study are available upon request from the corresponding author.

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
