# Peer review of "Preparation of AlON Powder by Carbothermal Reduction and Nitridation with Assisting by Silane Coupling Agent"

_materials, 2023, doi:10.3390/ma16041495_

Round 1

Reviewer 1 Report

Title: Preparation of AlON powder by carbothermal reduction and nitridation with assisting by silane coupling agent

·       The English may be improved, “The molecular of silane is mainly constituted by Si-base and C-base functional groups.”

·       Abstract:  Please insert the effect of time on the AlON powder, made at 1670 ℃. 

·       The objectives of the present study should be described including the processing parameters. 

·       Materials and Methods: please write about the grain size measurement. 

·       Please substantiate the carbon and oxygen maps using EDS.  The Figure caption may be exact, for instance: “Figure 1. The SEM images and correspond EDS mapping of Al2O3/C mixed powder with different contents of dispersants: (a) 0 wt%ï¼›(b) 2.5 wt%ï¼›(c) 5 wt%ï¼›(d) 7.5 wt%ï¼›(e) 10 wt%."        

·       Figure 10 can show the points.         

Reviewer 2 Report

Remarks: There are no principal essential remarks to the article. There are a number of comments on the design of the article:

1. The term "ALON" is a commercial name. The remaining chemicals in the article are given in the form of chemical formulas. Therefore, in the text of the article, you must either replace the commercial name with a chemical formula, or provide such a formula in section “2. Materials and Methods", for example, on line 75.

2. The first paragraph of the text (lines 220-224) from the section “Conclusions” should be moved to the end of the section “1. Introduction".

3. In the section “2. Materials and Methods" in line 76, it is necessary to give the synthesis temperature range under study.

Reviewer 3 Report

It is a nice round paper with only some points to clarify.

line 46: 'can be fluctuating'

line 52: what do you mean with 'relative investigation'?

line 92: looking at Fig.1 7.5wt% seems to be the one with the best uniformity in distribution of elements. Why did you choose 5%?

line 147: Fig. 5: also in Fig5(a) you have a certain peak at temperatures above 1200 °C. Therefore, I wouldn't say 'no noticeable change'. Please, refine your interpretation.

line 194: 'activity of powder'

line 212: 'lattice constants', 'holding times'

Reviewer 4 Report

The paper deals with an interesting subject of the effect of silane coupling agent on the synthesis of 63 AlON transparent ceramic powder. The paper is well prepared, but it needs a revision:

1. Abstract – please explain „an uniformity of mixed powder was the best” in the quantitative way.

2. What is the aim of the paper?

3. Ball milling process parameters and conditions should be stated.

4. Pleas describe the dimensions of powder – what is the distribution of powder particles diameters?

5. Fig 1 is invisible, what are  “C” and “O” photos, what is the scale for them?

6. Fig. 10 - please explain the change above 33 mol% AIN content.

7. Please comment the repeatability of achieved results.

8. Please summarize the achieved goals and add the info on the possible applications and future plans of the presented work.
